# Modeling of Packet Error Rate Distribution Based on Received Signal Strength Indications in OMNeT++ for Wake-Up Receivers

**DOI:** 10.3390/s23052394

**Published:** 2023-02-21

**Authors:** Mohamed Khalil Baazaoui, Ilef Ketata, Ahmed Fakhfakh, Faouzi Derbel

**Affiliations:** 1Department of Electrical Engineering and Information Technology, University of Applied Sciences, 04107 Leipzig, Germany; 2Laboratory of Science and Technologies of Information and Communication, National School of Electronic and Telecommunication of Sfax, Sfax 3000, Tunisia

**Keywords:** wake-up receiver, wireless sensor network, received signal strength, packet error rate, OMNeT++

## Abstract

Wireless sensor network (WSN) with energy-saving capabilities have drawn considerable attention in recent years, as they are the key for long-term monitoring and embedded applications. To improve the power efficiency of wireless sensor nodes, a wake-up technology was introduced in the research community. Such a device reduces the system’s energy consumption without affecting the latency. Thereby, the introduction of wake-up receiver (WuRx)-based technology has grown in several sectors. The use of WuRx in a real environment without consideration of physical environmental conditions, such as the reflection, refraction, and diffraction caused by different materials, that affect the reliability of the whole network. Indeed, the simulation of different protocols and scenarios under such circumstances is a success key for a reliable WSN. Simulating different scenarios is required to evaluate the proposed architecture before its deployment in a real-world environment. The contribution of this study emerges in the modeling of different link quality metrics, both hardware and software metrics that will be integrated into an objective modular network testbed in C++ (OMNeT++) discrete event simulator afterward are discussed, with the received signal strength indicator (RSSI) for the hardware metric case and the packet error rate (PER) for the software metric study case using WuRx based on a wake-up matcher and SPIRIT1 transceiver. The different behaviors of the two chips are modeled using machine learning (ML) regression to define parameters such as sensitivity and transition interval for the PER for both radio modules. The generated module was able to detect the variation in the PER distribution as a response in the real experiment output by implementing different analytical functions in the simulator.

## 1. Introduction

Nowadays, WSN plays a vital role in the modernization of daily life. The interconnection of smart wireless devices offers many easy services in various industrial, military, and civilian fields [1]. A sensor node can process the data collected from the physical environment, analyze them, and communicate them in transmitting or receiving mode. As a new data-collection technique, these networks have replaced wired sensor networks due to their advantages such as low cost and easy deployment. When the whole wireless network is based on a big number of distributed sensor nodes, it mainly deals with keeping the quality of the node’s communication as high as possible.

In line-of-sight or urban environments, WSN faces a lot of physical or environmental conditions, such as humidity, temperature, and vibration [2]. Among the important challenges facing WSN is the reliability and robustness to enable better wireless communication. Additionally, the network energy efficiency poses a challenge, as the sensor nodes are generally battery-powered and require autonomous operating for a long time. In recent research, the focus was on enhancing the capabilities of the hardware node for the optimization of WSN challenges [3].

Modern research has tended to improve wireless sensor networks (WSNs) in terms of latency, sensitivity, and energy efficiency. RF communication started with receiver-initiated media access (MAC) protocols, where the node has to listen to the channel; as a result, the radio frequency (RF) chip is powered to detect a potential beacon frame. This type of communication is not energy-efficient because idle listening is typically the primary source of energy consumption in receiver-initiated protocols [4]. Therefore, they switched to duty-cycled protocols where the receiver switches on for a period to listen for potential beacons. However, these protocols are limited in terms of latency.

The revolutionary invention in the world of sensor nodes is WuRx, which is an ultra-low-power receiver included in addition to the main transceiver as a solution for the idle listening issue. The WuRx ship is always in receiving mode, plays the role of detecting the wake-up call (WUC), identifying it, and then sending an interrupt to the main transceiver through the microcontroller unit (MCU), which remains in sleep mode until receiving the interrupt from the MCU. Figure 1 presents a block diagram of a WuRx, showing the interconnection between the WuRx, the main transceiver, and the MCU, with an RF switch in between to route RF signals through the transmission path.

The protocol scheme of the WuRx is simplified in Figure 2, indicating the transmission of the WUC when the receiver is in sleep mode. The data start transmission after identification of the WUC and activation of the main wireless chip.

Currently, two WuRx approaches are being pursued. The first one is classified in the application-specific integrated circuit (ASIC) WuRx [6], and the second one is based on commercialized off-the-shelf (COTS) WuRx [7,8].

The WuRx energy efficiency has switched investigations to the reliability of those receivers, because data-transferring performance relies on communication accuracy, which could be affected by external factors such as noise, interference, etc. One of the accurate methods used to determine the reliability of WuRx communication is to measure the PER according to the transmission power or the attenuation level, because this metric gives an overall idea about the sensitivity level of the sensor node [9]. Additionally, an RSSI that measures an approximate level of attenuation during the transmission [10] is used in some localization application techniques [11].

The use of a simulator is a key for the investigation of wide networks before implementing them in real situations. The simulation reduces the cost and time of implementation. Additionally, the simulators give researchers a general idea of WSN behavior that can be deployed in the future. Especially by working with WuRx, it is mandatory to test and simulate sensor nodes before network implementation because the power consumption of the wake-up nodes can be determined by calculations. However, the sensitivity of the receiver is an important parameter that should be estimated and verified.

A division of COTS WuRx is based on low frequency (LF) receivers, e.g., AS3933. Figure 3 shows the building blocks of an 868 MHz WuRx utilizing an envelope detector, an LF amplifier to increase the sensitivity of the radio chip, and an LF receiver for pattern recognition and interruption of the primary transceiver during its sleep mode. WuRx, being a recent representative of this low-frequency pattern matcher (LFPM), is the main focus of this article.

The increasing competition in the research field is demanding more accurate validation of studying and revealing real-world processing scenarios. Additionally, the wake-up receiver in the latest green technologies, which consumes only in the range of a few micro-watts, is a novelty. The main contribution of this study is materializing the analysis and the execution of an accurate model of a WuRx that includes two different radio chips. This model is integrated into an OMNeT++ discrete event simulator to perform the WuRx metrics, which opens the horizon for low-cost, short-term, future scenario simulations to investigate radio behavior.

This paper first presents a general overview of the reliability of an RF-WuRx that integrates an LFPM, a main 868 MHz transceiver, and an attenuation factor that affects the signals near the receiver’s antenna. Then, we describe the implementing software and hardware metrics in a simulator that enables the investigation of the reaction of the WuRx to different attenuation levels in different environmental conditions. We also provide test results.

The rest of this paper is organized as follows: An overview of the importance of link quality metrics in WSN reliability and a study of different simulators are presented in Section 2. In Section 3, an investigation of a PER using WuRx and a discussion of results are shown, while the implementation of different metrics and the simulation results in an OMNeT++ discrete event simulator are provided in Section 4. Finally, this paper is ended with a discussion and conclusions.

## 2. Related Work

### 2.1. Importance of Simulations for Wireless Sensor Networks

Nowadays, WSN requirements are rapidly growing due to their integration into different applications. WSN simulators are known as the most-used approach for protocol testing. Most researchers in the field of WSNs use simulators to predict a node’s behavior because simulators provide several advantages such as test period and flexibility. Every WSN application has to involve a conception and a design phase. After that, the test phase takes place. The simulation phase is relevant to help both designers and researchers obtain a general idea of the network. Many studies and surveys on available WSN simulation environments have been presented in the literature [12,13,14].

A study [13] shows that the design and optimization of specialized WSN platforms and communication protocols typically rely on simulation tools, which have been designed to explore and validate WSN systems before actual implementation and real-world deployment.

The ability to integrate real testing conditions into a simulator interface enhances the challenge of network accuracy and reliability. Additionally, the prediction of parameters that can affect communication can be taken into account, e.g., the variation in RSSI and PER degradation. This classifies simulation as the most-used evaluation technique in the wireless communication field due to its low-cost implementation and ease of use [12].

In [14], a statistical overview was performed on 403 papers to compare the most commonly used simulators from a research point of view. Figure 4 shows that the most-cited simulators are network simulator 2 (NS2),tiny OS simulator (TOSSIM), and OMNeT++ discrete event simulators.

The OMNeT++ discrete event simulator is among the top list of simulating tools that have drawn significant interest in research. Although the OMNeT++ discrete event simulator provides powerful and precise simulation frameworks, such as Simu5G, Veins, RinaSim, INET framework, etc. [15], the most-utilized framework is the INET framework, which provides protocols, agents, and other models for researchers and students working with communication networks. INET is especially useful when designing and validating protocols or exploring new or exotic scenarios.

In [16], the OMNeT++ discrete event simulator proved to be excellent WSN simulation software; its functions follow the needs of WSN simulation. Compared with NS2, the OMNeT++ discrete event simulator has better performance, but it also has some advantages over optimized network engineering tools (OPNET), which is expensive commercial software.

In [17], the OMNeT++ discrete event simulator was a great tool to obtain performance indicators such, as data extraction rate (DER), energy depleted by nodes, and loss rate of transferred packets in WSN.

In [18], the author modulated the impact of temperature and humidity on the RSSI in the OMNeT++ discrete event simulator using Panstamp AVR2 wireless sensor nodes. According to the author, OMNeT++ was chosen as a simulator because of its hierarchical architecture that simplifies the integration of new modules; additionally, it is powerful graphical user interface (GUI).

Table 1 classes the three most-mentioned software cited according to [14] and makes a comparison for different specifications.

In [16], a comparison was made between the well-known simulators, including NS2, OPNET, and the OMNeT++ discrete event simulator, which showed that the OMNeT++ discrete event simulator has better performance in terms of large-scale WSN simulation than the other simulators.

Because the OMNeT++ discrete event simulator integrates more wireless medium and energetic models, the simulations of a WSN are more realistic than those of other simulators. Additionally, the hierarchical structure of the models makes it one of the best choices for modeling various experiences in real life. The GUI of the OMNeT++ discrete event simulator is based on the Qtenv environment, which supports interactive simulation execution, which we cannot find in other simulators. However, the OMNeT++ discrete event simulator has some limitations: it lacks observation of some metrics, especially for new WSN protocols such as WuRx. The main objective of this study was the integration of those metrics after specific measurements.

### 2.2. State of the Art of Link Quality Metrics of Variation in Wireless Sensor Networks

To define the reliability of communication in wireless networks, researchers must rely on many link quality metrics, most of which are illustrated in two main categories: software- and hardware-based metrics.

Hardware metrics are obtained using the hardware of the transceivers that estimate the received signal level or noise level using different algorithms after listening to the operating channel.

Software metrics are acquired using the statistics for the reception error rate after a definite time interval or a certain amount of delivered beacons.

In WSNs, the quality of communication between sensors is evaluated using link cost metrics, also known as link estimators [19]. Figure 5 demonstrates the organization of different metrics.

### 2.3. State of the Art of Software Metrics in the Estimation of Link Quality in Wireless Sensor Networks

Because the variation in the PER defines the sensitivity of the wireless sensor node, it plays an important role in the robustness of communication, energy efficiency, and routing protocols.

The PER is the transmission error, where the data transmission unit receives erroneous information from one of its sources. This might happen due to interference, attenuation, or hardware failure. It can also result from unreliable data sources. Packet errors are measured in percentage as the number of packets lost divided by the total number of sent packets. However, the packet reception ratio (PRR) calculates the reception ratio in which the received packet is correct.
(1)$$PRR=1-PER=\frac{n_{received}}{n_{sent}}$$
where $n_{received}$ is the total number of successfully received packets by the receiver, and $n_{sent}$ is the total number of the sent packets by the transmitter.

The industrial environment is a challenging field for sensor node deployment because the link quality is constantly affected by electromagnetic waves, noise, and interference from different technologies. The estimation of link quality is a needed process to define network reliability, such as in [20], where a PRR was used in addition to the average of the signal to noise ratio (SNR) and link quality indicator (LQI).

The PER metric is an essential software-based metric in wireless networks, especially in sensor nodes and mobile ad hoc networks, because the reliability of packet reception ratio has a fundamental impact on the efficiency of wireless communication [19].

Recently, PER has drawn much attention due to its advantages: simplicity and efficiency. In [21], a theoretical bit error rate (BER) model of IEEE802.15.4 and a PRR estimation approach were proposed using a simulation platform (Matlab 2013a).

Researchers [22] studied the relationship between system reliability and transmission range using different metrics such as the PRR and symbol error rate.

In [9], the performance of WuRx was discussed based on two LF amplifier approaches according to the sensitivity, power consumption, and PER. An indoor experiment was carried out inside a building in different rooms and on different floors.

An author [23] presented sender-side software-based link-quality estimator metrics, the link quality level (LQL) and the expected transmission count (ETX), for approximating link quality by counting the ratio of the number of transmitted and retransmitted packets to the number of successfully received packets to build a more reliable routing based on the link quality.

In [24], the researchers combined both hardware- and software-based metrics and used the PRR to overcome the lost packet information problem and improve the accuracy rate for evaluating a link quality in a WSN.

In [25], a combination of PER, SNR, and LQI using IEEE 802.15.4-compliant transceivers (2.4 GHz Chipcon CC2420 radio) was used to provide a real-time estimation of the link quality between nodes.

The results of the previous studies have shown that the PER is strongly correlated with the received signal power, path loss, SNR, etc. Software-based metrics can more precisely estimate link quality than hardware-based metrics. However, using only software-based metrics to evaluate a link quality cannot capture link quality changes in real-time [25].

### 2.4. State of the Art of Hardware Metrics in the Estimation of Link Quality in Wireless Sensor Networks

The RSSI of RF signals in WSNs is intended for various applications. The RSSI is a low-cost solution for distance estimation that replaces the global positioning system (GPS) in WSNs, which makes it a practical choice for localization schemes [11,26]. The RSSI is also used as a hardware metric to estimate the link quality in wireless communication, as in [20], who used the RSSI in addition to the LQI for determining the link quality by using a Kalman filter and fuzzy logic. In [27], the RSSI and LQI were selected as estimation parameters to build a link quality estimation model based on support vector machine (SVM) with multi-class classification.

In the free space, non-obstructive environment, the nature of electromagnetic wave propagation abruptly attenuates signal power near the transmitter and yields much less attenuation at further distances. This is described by the Friis Equation [28], which is directly derived from fundamental theory.
(2)$$P_{R}=\frac{P_{T}G_{T}G_{R}\lambda^{2}}{{\left(4\pi d\right)}^{2}}$$
where $P_{R}$ is the power at the receiving antenna; $P_{T}$ is the output power of the transmitter; $G_{T}$ is the transmitter’s antenna gain; $G_{R}$ is the receiver’s antenna gain; $\lambda =\frac{c}{f}$ is wavelength; *d* is the distance between the antennas, as shown in Figure 6.

The Friis equation can be translated into numerous models. Because signal attenuation is usually linked to various environmental conditions, choosing the signal propagation model that is the most appropriate for the current environment is fundamental to ensuring the accuracy of signal power computation in a simulator. The shadowing model is widely used to represent wireless signal propagation loss, which translates into the following equation: (3)$${\left.P_{r}\right|}_{dBm}\left(d\right)={\left.P_{r}\right|}_{dBm}\left(d_{0}\right)-{\left.10\right|}_{dB}\times n\times log_{10}\left(\frac{d}{d_{0}}\right)+{\left.X_{\sigma }\right|}_{dB}$$
where ${\left.P_{r}\right|}_{dBm}\left(d\right)$ is the received signal power at the real distance; ${\left.P_{r}\right|}_{dBm}\left(d_{0}\right)$ is the received signal power at a distance of reference; *d* is the distance between the antennae; $d_{0}$ is the reference distance between the antennae; *n* is the attenuation factor; ${\left.X_{\sigma }\right|}_{dB}$ is the zero-mean Gaussian distributed random variable with a standard deviation of σ.

In most cases, RSSI-based ranging algorithms employ the shadowing model, which is nowadays used in localization-based algorithms, which use the trilateration method. This method needs the presence of three nonlinear reference points that are represented by three transmitters. A node applies RSSI measurements to approach its distances from the received packets [29].

In real circumstances, many external factors can attenuate the wave signals near the receiver, such as the reflection, diffraction, or absorption caused by a different type of obstacle; the physical environment such as wind, rain, temperature, lightning; and so on.

WSN applications have been manipulated into various domains in indoor or outdoor scenarios. Different researchers have studied the impact of various environmental conditions on the LQI and RSSI. In [30], a study case was made to investigate the link quality and the received signal strength using the TelosB nodes of a WSN in a metal marine environment, where 18 nodes were distributed in a complex metallic environment composed of freight containers, engines, and different materials that cause the attenuation of signals.

In [31], the impact of temperature on the RSSI and LQI were investigated in an oil refinery using CC2420 radio ships. The experimental results showed that temperature had a significant influence on signal strength and link quality.

In [32], the impact of humidity and temperature on the RSSI in indoor WSNs was explored; the 868 MHz Panstamp NRG 2.0 wireless modules were disposed of, where different distances showed various values of the interaction of the RSSI according to the humidity and temperature.

In [33], the effects of ambient temperature and humidity on the radio signal strength of Atmel ZigBit 2.4 GHz wireless modules were explored in an outdoor WSN. The experimental results demonstrated that changes in weather conditions influenced the received signal strength. The temperature seemed to have a more significant negative impact on the signal strength in general, while high relative humidity may have had some effect on it, particularly below $0{\;}^{\circ }C$.

An experiment demonstrated a linear decrease of 8 dB in signal strength when the temperature rose from 25 °C to 65 °C using a TI CC2420 radio chip on a Tmote Sky node. They also showed the implications of the experiment on different communication ranges [34].

In a study [35], a sensor network comprising 16 TelosB sensor nodes was manipulated in outdoor measurement over a year, revealing that the RSSI had a negative correlation with temperature.

In [9], the variation in RSSI was investigated for different rooms and floors of a building using different hardware technologies for the amplifier in a WuRx, e.g., bipolar junction transistor (BJT) and transimpedance amplifier (TIA) technologies. The highest RSSI was achieved when both WuRx and wake-up transmitter (WuTx) were on the same floor and near each other. At a distance of more than 4 m, the RSSI abruptly dropped.

In [36], an experiment to measure the RSSI using a CC2420 2.4 GHz radio chip was was conducted in outdoor and indoor environments within the line of sight. The experiment curve showed that the deviation in the RSSI value was different from one environment to another. In indoor environments, the noise is higher due to the presence of obstacles with a higher probability of the reflection of the waves.

The physical study of the software and hardware link quality metrics in this article proves the important role of network reliability optimization in many studies. Additionally, multiple strategies can be adopted in future work for more investigation into the scalability process of networks, such as [37], who proposed a simulation for a proposed approach for improved routing, or the [38], who cited the optimization of some real traffic or communication networks to control congestion.

### 2.5. Discussion of Previous Studies

Many studies have been performed on link quality software based on wireless communication, such as in [9,21,22], but none of the previous researchers have drawn attention to the modeling or integration of link quality estimators of WuRx technology in a simulator, where the simulator has an important role of reducing the time and cost of the integration of different sensor nodes in real applications.

Many researchers have proved the impact of the external environment on the RSSI [31,32,33,34]. One author [9] studied the variation in the RSSI in a WuRx in an indoor environment; in [36], different noise figures of the RSSI metrics were shown in different experiments for indoor and outdoor environments; however, to the best of our knowledge, none of the wireless simulators take into account those impacts.

All the proposed studies focused on the physical implementation of sensor nodes with limited results that may be insufficient for analyzing and model interpretation in relation to the proposed problem. As described before, dealing with an infinite number of simulations may produce better results for the node’s behavior, which was implemented in this study based on real measurements.

## 3. Investigation of the Packet Error Rate Variation Based on Wake-Up Receiver and SPIRIT1 Transceiver

### 3.1. Experiment Setup for the Investigation of the Impact of Attenuation on the Packet Error Rate

Experiments for PER investigation are widely used for the sensitivity measurement of the sensor node, such as in [39], in which the behavior of the AS3933 LFPM was measured using an RF generator to generate the RF wake-up packet (Wupt) pattern, which was directly fed into the hardware to analyze WuRx performance.

In this work, the hardware of the WuRx node was composed of a SPIRIT1 transceiver that sent the WUC signals to the COTS WuRx with an integrated LFPM AS3933. Both the radio and COTS WuRx hardware were attached to a Texas Instrument microcontroller MSP430G2553, as shown in Figure 7.

To control the attenuation level, a digital RF attenuator ADAURA was included between the WuTx and the WuRx. This digital attenuator was tested with the continuous wave (CW) mode of the SPIRIT1 transceiver that was attached to a spectrum analyzer, as shown in Figure 8, and it was accurate with a ±0.25dB maximum variation. The configuration specifications of the attenuator are shown in Table 2.

Two experiments were launched to test the PER variation in the WuRx and the SPIRIT1 radio transceiver. Because different radio ships were used in the experiment, the link budget varied from one hardware to another; the maximum attenuation range of the RF digital attenuator was 95dB.

SPIRIT1 communication profiles use frequency shift key (FSK) modulation with 38.4 kbits${}^{-1}$. According to [40], the sensitivity limit of SPIRIT1 when the integrated switched mode power supply (SMPS) is off is around −109 dBm. To reach this level of sensitivity, a drop in the transmission power was configured in the SPIRIT1 radio ship, see Table 2. The minimum result link budget was $LB_{min}$ = −114 dBm.

To reach the lowest power consumption of the LFPM AS3933, an internal RC oscillator was used with an 18.72 kHz frequency; typical AS3933 sensitivity is $80\;\mu V_{RMS}$. Theoretically, the sensitivity level can reach −65dBm; according to Table 2, $LB_{min}=-69\;dBm$.

#### 3.1.1. Packet Error Rate Measurement of Wake-Up Receiver

The measurement setup was conceived using two nodes, WuRx and WuTx, with peer-to-peer communication. The SPIRIT1 transceiver was programmed to send Wupt with a power of 11dBm and to wait for an acknowledgment from the receiver. The WuTx sent 100 consecutive packets and waited for the acknowledgment for each packet. Finally, the number of received and lost packets was measured.

The transmitted Wupt was an 868 MHzon-off keying (OOK) signal at a specific data rate to match the operative frequency of the LF AS3933 receiver after passing by the envelope detector. We used a Schottky diode in the envelope detector to convert the RF signal into LF signals.

When the AS3933 validated the Wupt passed by the envelope detector, it sent an interrupt to the MSP430G2553 to activate the main transceiver and turn it into Idle reception mode, waiting for the data packet. When the data were received, the SPIRIT1 transceiver sent back an acknowledgment (ACK) to inform the transmitter that the packet had arrived. A sequence diagram shows the communication protocol process in Figure 9.

The experiment was repeated several times starting from 53 dB to 80 dB of attenuation of attenuation with a step of 0.5 dB. The count of the lost packets is plotted in Figure 10 to demonstrate the impact of attenuation on the PER for AS3933.

When the attenuation went from 66 dB to 69 dB, the PER rose to 100%. That showed the limit of the sensitivity in the WuRx. At this point, the transmission power was conserved, and the attenuation level was 66dB. The SPIRIT1 transceiver was attached to a spectrum analyzer to measure the received output power, which was −58dBm. We had 11dBm transmission power; the output level was meant to be −55dBm, but the 3 dB here was the was the result of the losses due to the transmission losses in the cable.

#### 3.1.2. Packet Error Rate Measurement of SPIRIT1 Transceiver

To measure the reliability of the SPIRIT1 transceiver, a modification to the protocol was made. Because the WuRx was not tested, the SPIRIT1 transceiver directly sent the data packet sequence to the receiver without a WUC.

The receiver was always in idle mode, listening to the channel for arriving packets. When the data were successfully transmitted, an acknowledgment (ACK) was sent back to the transmitter. At the end, the receiver counted the sum of successfully received packets. The sequence diagram in Figure 11 describes the communication protocol used to test the PER SPIRIT1 transceiver.

The same setup for the previous experiment in Section 3.1.1 was used. The transmission output power was programmed to be −34dBm to reach the sensitivity limit level of the SPIRIT1 transceiver. The experiment was repeated several times, starting from 62dB to the 90dB level of attenuation, with a step size of 0.5dB. The count of the lost packets is plotted in Figure 12 to demonstrate the impact of attenuation on the PER using the SPIRIT1 transceiver.

When the attenuation rose from 70dB to 75dB, the PER changed from 6% to 100%. This showed the limit of the sensitivity of the SPIRIT1 transceiver; at this point, the transmission power was preserved, and the level of attenuation was 70dB. The SPIRIT1 transceiver was attached to the spectrum analyzer to obtain the received output power, which was −107dBm.

### 3.2. Theoretical Model of the Packet Error Rate Variation for Wake-Up Receiver Based on AS3933 and SPIRIT1 Transceiver

After collecting the measurement data, and inspecting, transforming, and cleansing the data, the second stage was to create a data model for the information system by applying certain formal techniques to extract the system behavior. Using the collected data explained in the previous section, we then theoretically modeled the PER, using a ML algorithm that uses nonlinear regression mathematical models based on supervised learning.

A study [21] proposed a theoretical model based on PRR in IEEE802.15.4 2.4 GHz technology using the Q-function, which is the tail distribution function of the standard normal distribution.

In this study, the model was built using a nonlinear regression function known as the error function, the error function—also called the Gauss error function—is a special (non-elementary) sigmoid function that often occurs in probability, statistics, and partial differential equations. The representation of the function is shown as follows: (4)$$erf\left(x\right)=\frac{1}{\sqrt{2\pi }}\int_{0}^{x}e^{-\frac{t^{2}}{2}}dt$$

Because different curves are applied for various hardware and the needs of the input parameters for the regression required us to define an error function applicable to the collected data, the input parameters were:The transition interval, which is the interval of attenuation that passes the PER from 0% to 100% or the inverse.The sensitivity level, which is the center of the transition interval to define the sensitivity level of the hardware.
(5)$$PER\left({\left.RSSI\right|}_{dBm}\right)=\frac{1}{\sqrt{2\pi }}\int_{0}^{\frac{({\left.RSSI\right|}_{dBm})-d}{l}}e^{-\frac{t^{2}}{2}}dt+\frac{1}{2}$$

This equation can be applied to the extracted PER set of data for both the transceiver and WuRx, with *d* being the sensitivity level and *l* being the transition interval multiplied by 4.

The built-in ML algorithm applied nonlinear regression and used the ordinary least square (OLS) method in order to minimize the deviation between the estimated and real curves.

Figure 13 shows the estimated modeling function for the PER in the SPIRIT1 transceiver according to the RSSI. The calculation result showed a −111.27dBm sensitivity and an around 6.9dB transition interval. Compared with the demonstrated measurement shown in Section 3.1.2 the rise in the PER started at −107dBm. Meanwhile, in the model, the change started from −107.82dBm.

Figure 14 shows the estimated modeling function for the PER in AS3933 WuRx according to the RSSI. The calculation result showed a −59.55dBm sensitivity and an around 3.01dB transition interval. Compared with the demonstrated measurement shown in Section 3.1.1, the rise in the PER started at −58dBm; meanwhile, in the model, the change started from −58.5dBm.

The modeling purpose was to mimic real WuRx behavior. The fact is that the curves were not as smooth as in Figure 14 for the AS3933 and as in Figure 13 for the SPIRIT1 transceiver. The proposed solution was to add a noise figure to the curves of both the SPIRIT1 transceiver and AS3933, a distribution that minimized the OLS.

A quantile−quantile plot is a visual tool that helps with evaluating if a set of data plausibly come from some theoretical distribution such as a normal or exponential distribution. In this study, we ran a Q-Q plot for both SPIRIT1 and WuRx radios that took our residuals and compared them with the estimated curves.

We noticed that in Figure 15 and Figure 16, the samples are consistently on the edges of the graph, and the data set is not intensive when we considered the transition interval of the radios. For that purpose, the chosen distribution for this study was the Weibull distribution due to its ability to provide reasonably accurate analysis estimation, even for a small set of data. The Weibull distribution is a two-parameter family of continuous probability distributions over a set of positive real numbers. Depending on its two parameters, it resembles a normal distribution or an asymmetric distribution, such as an exponential distribution. The Weibull function is written as follows:
(6)$$f\left(x;\lambda ,k\right)=\left\{\begin{array}{cc} \frac{k}{\lambda }{\left(\frac{x}{\lambda }\right)}^{k-1}e^{-{\left(x/\lambda \right)}^{k}} & x\geq 0\\ 0 & x<0 \end{array}\right.$$

The square distances of the nonlinear function of the PER were extracted, and the Weibull function was applied over the noise figure using kλ and a multiplication factor *m* as input parameters for both the WuRx based on the AS3933 and SPIRIT1 main transceiver. The model was used as a filter to minimize the residuals through multiple stages after applying the erf(x) function, as shown in the block diagram in Figure 17.

Where s(t) is the original signal; s1(t) is the signal after applying the error function and obtaining the sensitivity and the transition interval; o1(t) is the residual output from the original signal; o(t) is the output signal with the noise distribution.

This function was implemented, and the results are shown after the simulation in the OMNeT++ discrete event simulator in Section 4.1.

## 4. Implementation of the Experiment Results in OMNeT++ Discrete Event Simulator

### 4.1. Implementation of Packet Error Rate Distribution in OMNeT++ Discrete Event Simulator for the Study Results

The aim of this work was to model the link quality estimators in the OMNeT++ discrete event simulator. The PER in OMNeT++ discrete event simulator varies according to the obtained power level at the receiver’s antenna, which is given by the RSSI. In addition to the characteristics of the used hardware, as a result of the measurements in Section 3.1.1 and Section 3.1.2, different levels of sensitivity and other values for the PER to rise from 0% to 100% were given.

The OMNeT++ discrete event simulator has a predefined error model, which is located in the physical layer of the INET framework. In the packet-level package, the model contains different error models such as BER, sequence error rate (SER), and PER. This model is static, lacks inputs, and is limited. It is based on the variation in the distance in 802.11g wireless networks.

The PER function was implemented in the physical layer of the INET framework in the APSK error model. This module is based on the RSSI module, in addition to the sensitivity and the transition interval parameters that define the radio chip. The sensitivity and the transition interval are initialized by the user using the initialisation (INI) file. An editor considers all supported configurations to different module options and offers them in several forms. The module is described by the inheritance diagram in Figure 18.

The computePacketErrorRateBrssi application protocol interface (API) includes the error function modeled in Section 3.2, with the Weibull distribution that was included in the transition interval when the PER passed from 100% to 0%. The PER calculation was registered during the simulation in the receiver part as a signal message. The OMNeT++ discrete event simulator has a hierarchical architecture: it connects different modules, and those modules communicate through signals to send it to the result-recording module. The results are saved as a vector.

A simulation scenario was built in the OMNeT++ discrete event simulator, where two WuRxs were becoming closer to each other. The path-loss attenuation decreased, resulting in the RSSI built-in function increasing and the PER dropping from 100% to 0%.

Figure 19 gives the simulation result on the wake-up radio chip based on AS3933 after entering the according sensitivity and transition interval, which are provided by the nonlinear regression error function in Section 3.2: −59.55dBm sensitivity and around 3.01dB transition interval. It is clear that the distribution had no significant impact on the PER curve, as shown in Figure 20. The obtained result for the WuRx based on the LFPM was due to the shorter transition interval of the AS3933 WuRx 3dB compared with that of the SPIRIT1 transceiver, which is almost 7dB.

### 4.2. Implementation of the Received Signal Strength Indication in OMNeT++ Discrete Event Simulator

The RSSI is the power level of the signal received at a device’s antenna. The RSSI is determined by the transmission power, the distance between the transmitter and the receiver, and the radio environmental field, which can disturb RF wave signals beneath the receiver’s antenna, such as reflection that bounces the waves in various directions. The refraction of waves involves a change in the direction of the waves as they pass from one medium to another; diffraction, which scatters the wave and spreads it around the line of sight; or conditions such as weather changes.

RSSI-based location techniques demonstrate the relationship between the RSSI and the distance in wireless sensor networks, and this relation is computed according to the following equation: (7)$${\left.RSSI\right|}_{dBm}={\left.RSSI_{0}\right|}_{dBm}-{\left.10\right|}_{dB}n\times log\left(\frac{d}{d_{0}}\right)+{\left.X_{\sigma }\right|}_{dB}$$
where $RSSI_{0}$ indicates the RSSI when the reference distance is $d_{0}$;*n* indicates the path loss index in a specific environment; $X_{\sigma }$ is in dB, being a distribution with σ-level standard deviation. The more noisy the environment, the greater the deviation in the signal, which becomes more complex to predict.

The RSSI module was implemented under the physical layer into the compound module of the narrow-band receiver, as shown in Figure 21. Additionally, it was computed with each received packet, encapsulated as a signal tag to have the possibility of sending it to different modules, recording it, and showing the output results in a graph, as in the PER module.

The RSSI module used the RSSI-based location technique in Equation (Equation 7). The distance was obtained from different positions of the nodes in the simulator. The RSSI reference as the distance reference, the attenuation coefficient *n*, and the distribution level $X_{\sigma }$ were entered by the user as parameters before the simulation to obtain the system response in different environments. The diagram block Figure 22 shows the module location and its initiation parameters. The RSSbase module was created to collect information from different modules, such as different node positions.

A simulation process was built in the OMNeT++ discrete event simulator, where the distance between the WuRx nodes is increasing. The receiver node record the RSSI values with each received packet. Four scenarios were included to study different cases of noise figures, where the standard deviation varied from 0.5dB, as shown in Figure 23a, to 1.5dB, as shown in Figure 23b. As it was mentioned before, different environments are characterized by various noises: the more persistent the noise, the higher value of the deviation. As shown in Figure 23c,d, another log-normal distribution was built in the simulator with different widths, 0.5dB and 1dB, to better show how RSSI changes in fading channels.

## 5. Summary

Summarizing the overall achievements of the proposed method, the OMNeT++ discrete event simulator was chosen over the other WSN simulators classified in Table 1 [14]. This simulator has a rich framework, and it was built with hierarchical integrated modules that enable the integration of new modules. In addition to the computation power, the performance of the OMNeT++ discrete event simulator was enhanced in terms of large-scale WSNs.

The choice of WuRx was made due to its ultra-low-power receivers. This new technology has been recently widely used and none of the simulators allow the integration of new modules for investigation of the reliability of networks. The importance of different link quality estimators was proven for the study of the reliability of WSNs, and the combination between software-based metrics and hardware-based metrics can increase the precision of the link quality estimation [25].

The integration of different radio chips in the same node leads to a differentiation in the characteristics of the two radio modules. These characteristics were taken into account in the response of the WuRx to a different level of attenuation. Moreover, the behavior of the main radio transceiver SPIRIT1 was investigated. Different analytical functions and the integration process of the PER metric in the OMNeT++ discrete event simulator were modeled. The integration of the RSSI module in OMNeT++ proved that different noise figures are output in the simulation for the investigation of WSN based WuRx technology.

## 6. Conclusions

In this study, the adoption of link quality metrics in the OMNeT++ discrete event simulator was performed, using the new technologies of WuRx nodes in WSNs. WuRx nodes were chosen due to their energy efficiency and the absence of their consideration in the simulators, where simulation is important to reveal the response of new WuRx technologies in many different fields.

The modeling of the PER was achieved through the testing and measurement of the impact of the attenuation on two radio chips, the SPIRIT1 transceiver and the COTS WuRx based on the LFPM AS3933.

Both radios operate in the 868 MHz central frequency, and a specific data rate was considered for matching the LFPM AS3933 central operating frequency, which is 18.72 kHz with a specific address.

Different levels of attenuation between the transmitter and the receiver were applied, due to the distinctions in the link budget of different radios. The main radio SPIRIT1 transceiver reached a −111.27dBm sensitivity. The WuRx based on the LFPM AS3933 reached a −59.55dBm sensitivity. The investigation showed the presence of a larger deviation in the SPIRIT1 transceiver than in the WuRx based on the LFPM AS3933, which split from the error function regression model, which was handled by the Weibull distribution. The distribution parameters were built using a ML algorithm that used nonlinear regression to generate a precise curve compared to the real experiment output. In the end, a different noise level was applied to determine the output signal at the end of the simulation.

Finally, an accurate simulated model in OMNeT++ revealed the real reaction of the WuRx to different attenuation levels. This real-life simulation is a predeployment WSN-based WuRx for indoor applications. Many energy-efficient algorithms routing protocols supporting scalable WSN-based WuRx will be a matter of investigation in future works.

## Figures and Tables

**Figure 1 sensors-23-02394-f001:**
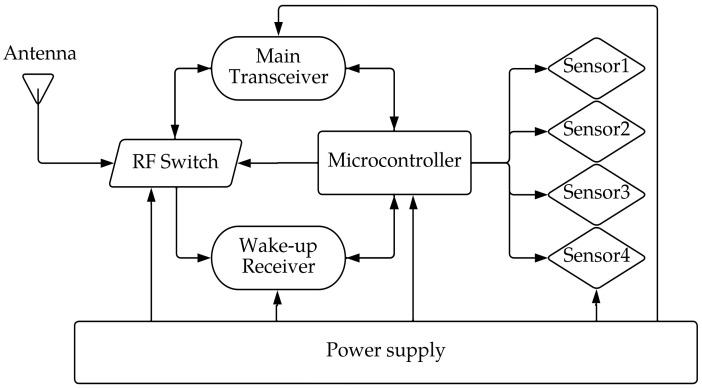
Wake-up receiver node block diagram according to [5].

**Figure 2 sensors-23-02394-f002:**
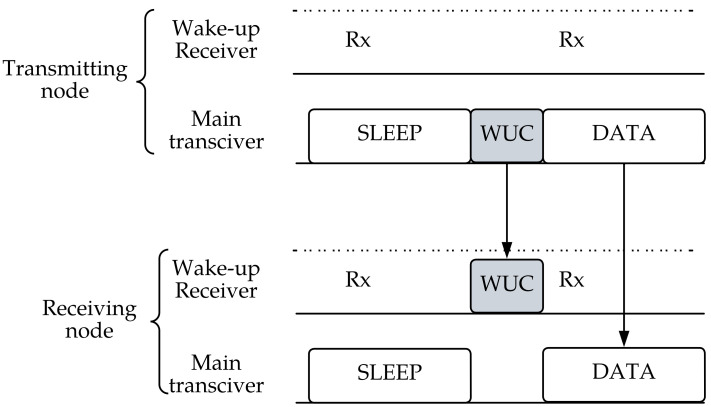
An example of a wake-up receiver-based protocol where the transmitter initiates the communication by sending a wake-up call, followed by data transmission.

**Figure 3 sensors-23-02394-f003:**
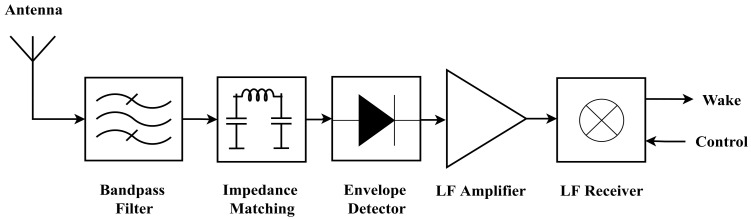
Building blocks of a typical commercial off-the-shelf WuRx with passive envelope detector and low-frequency receiver.

**Figure 4 sensors-23-02394-f004:**
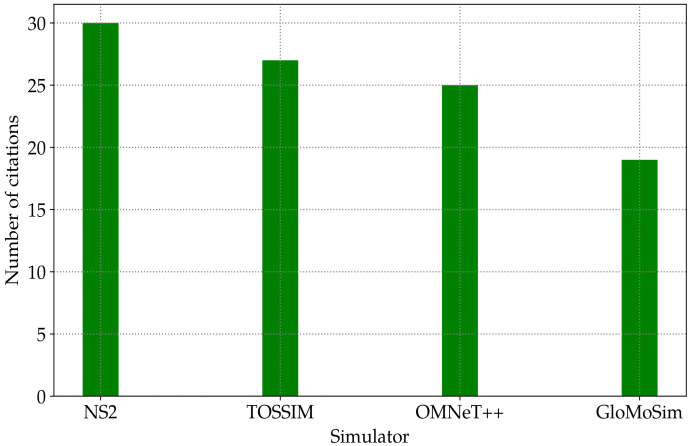
Number of citations that present different wireless sensor network simulators according to [14].

**Figure 5 sensors-23-02394-f005:**
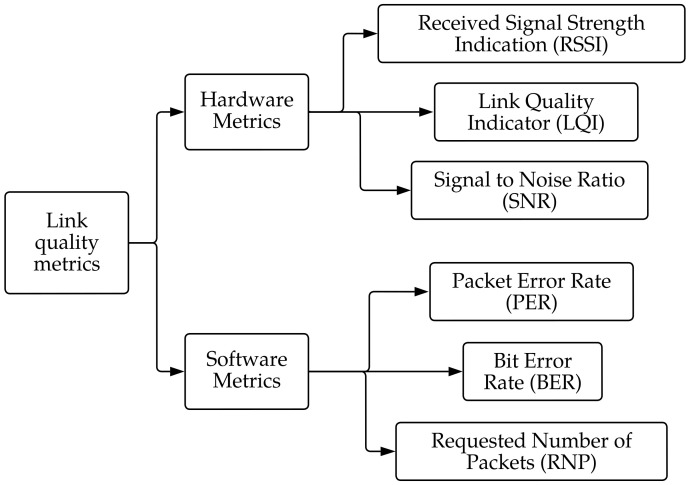
Organization of different link quality estimators according to [19].

**Figure 6 sensors-23-02394-f006:**
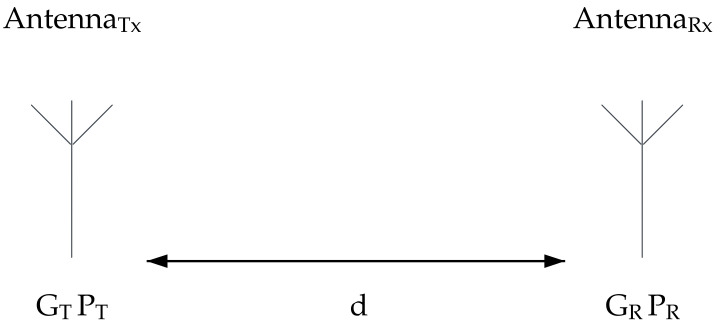
Free space Friisequation parameters.

**Figure 7 sensors-23-02394-f007:**
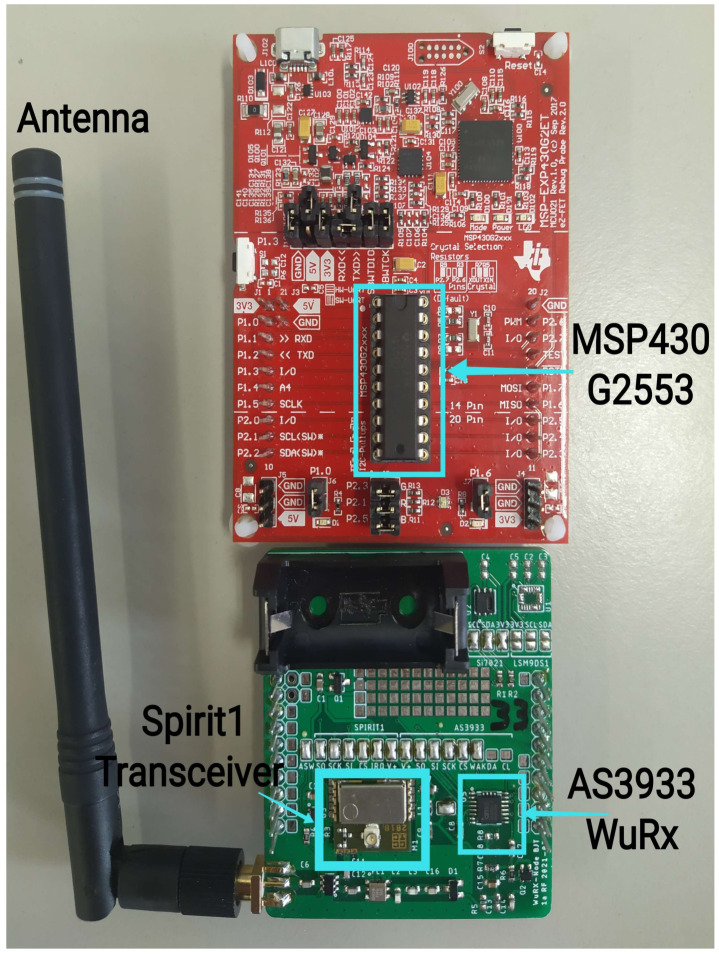
Photograph of WuRx node RF shield on the bottom and MSP430 launchpad on the top.

**Figure 8 sensors-23-02394-f008:**
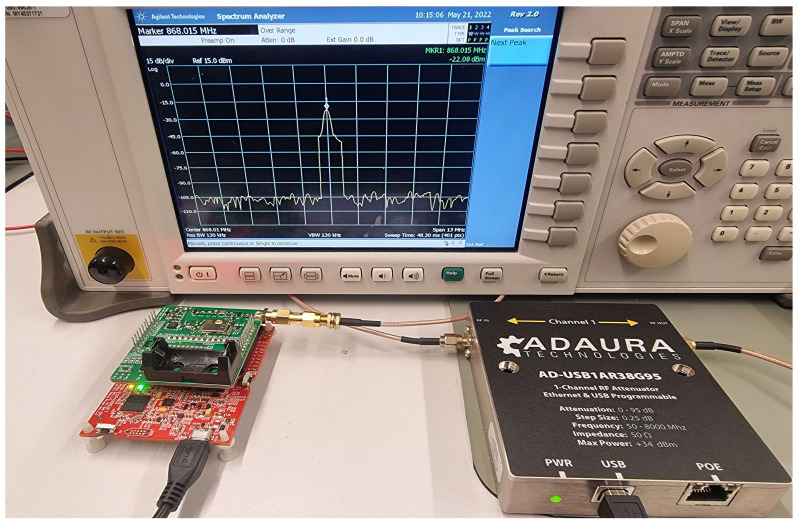
Attenuation accuracy test using continuous wave mode in SPIRIT1 transceiver.

**Figure 9 sensors-23-02394-f009:**
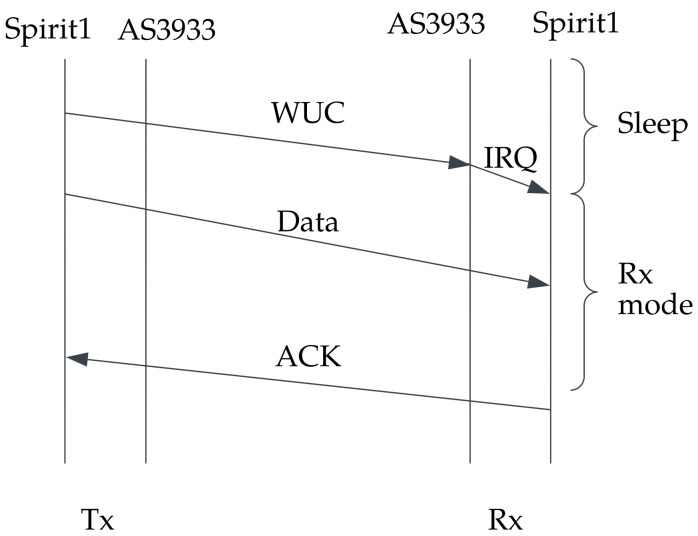
Communication sequence diagram.

**Figure 10 sensors-23-02394-f010:**
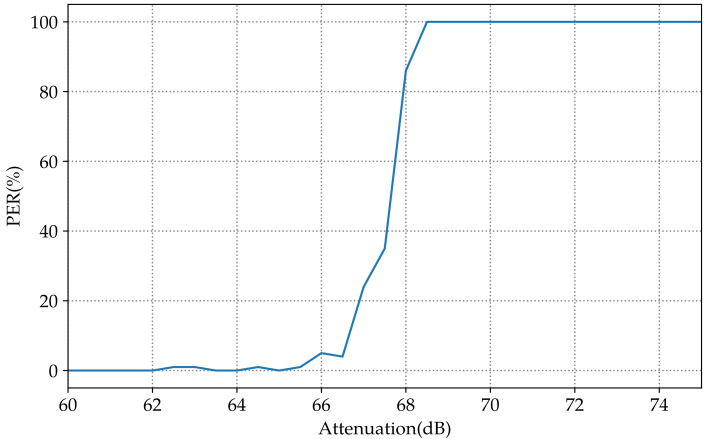
Variation in the PER according to the attenuation factor using WuRx communication.

**Figure 11 sensors-23-02394-f011:**
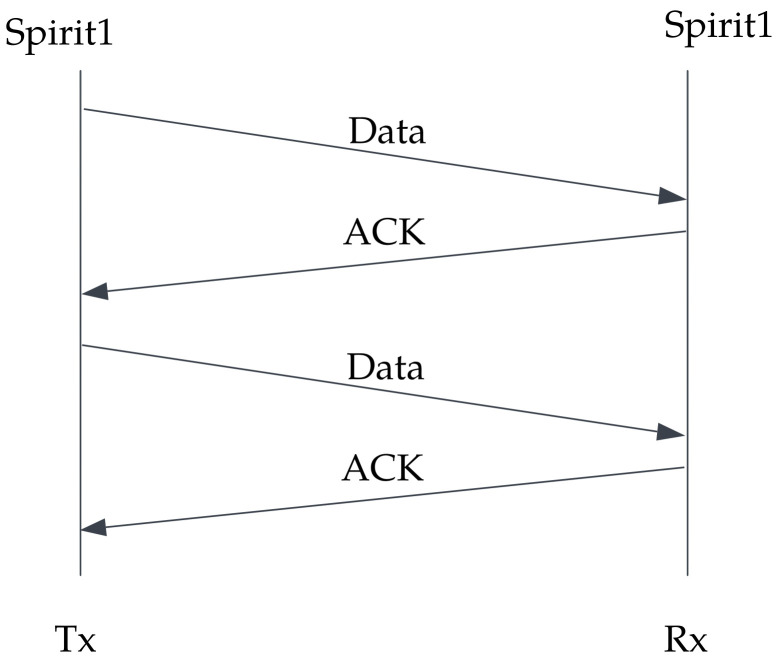
Communication sequence diagram using SPIRIT1 transceiver.

**Figure 12 sensors-23-02394-f012:**
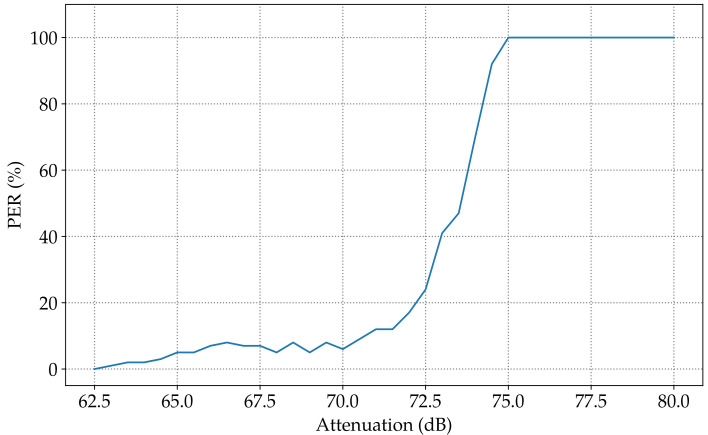
Variation in the PER according to the attenuation factor using SPIRIT1 communication.

**Figure 13 sensors-23-02394-f013:**
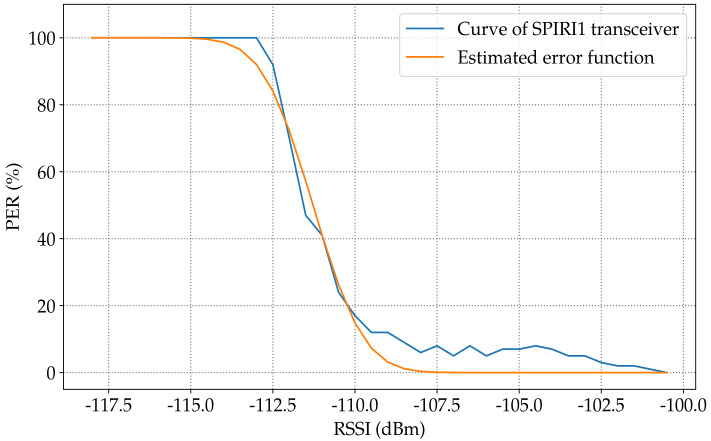
Estimation of the PER curve according to the RSSI in the SPIRIT1 transceiver.

**Figure 14 sensors-23-02394-f014:**
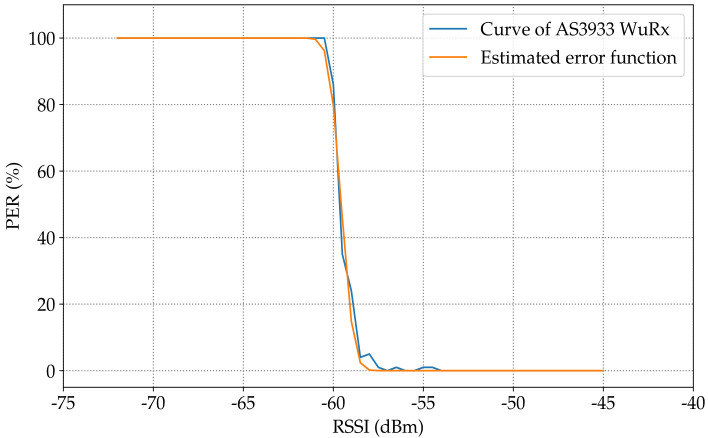
Estimation of the PER curve according to the RSSI in AS3933 WuRX.

**Figure 15 sensors-23-02394-f015:**
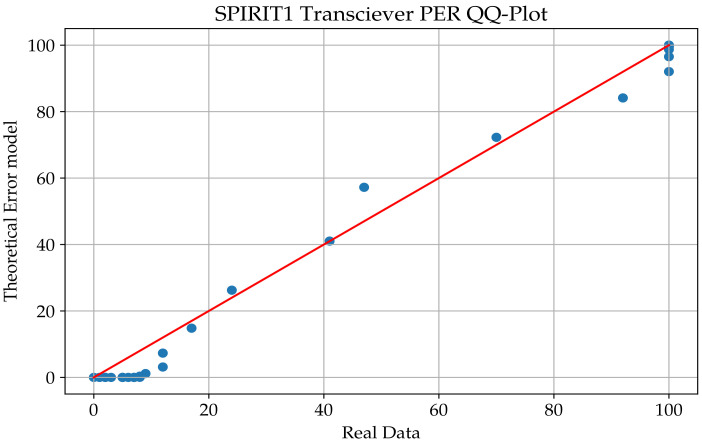
Q-Q plot of the SPIRIT1 transceiver.

**Figure 16 sensors-23-02394-f016:**
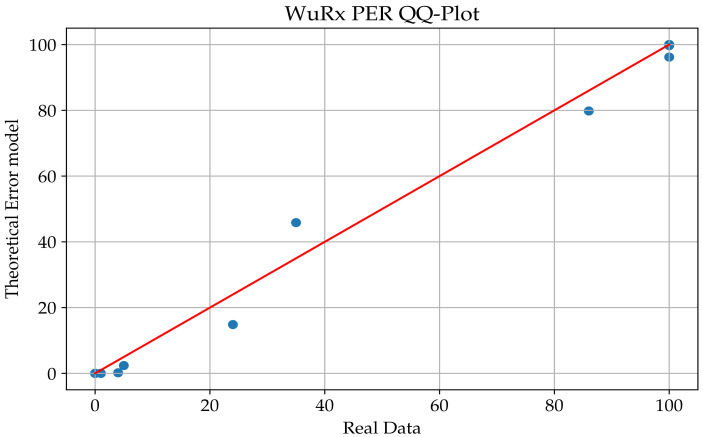
Q-Q plot of the WuRx.

**Figure 17 sensors-23-02394-f017:**
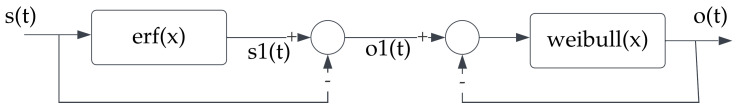
Noise distribution modeling block diagram using multiple stages of Weibull function.

**Figure 18 sensors-23-02394-f018:**
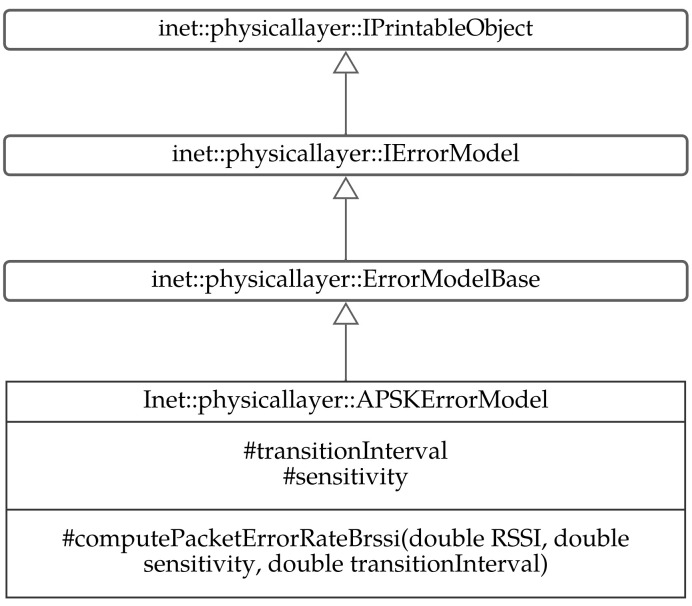
PER module in OMNeT++ discrete event simulator.

**Figure 19 sensors-23-02394-f019:**
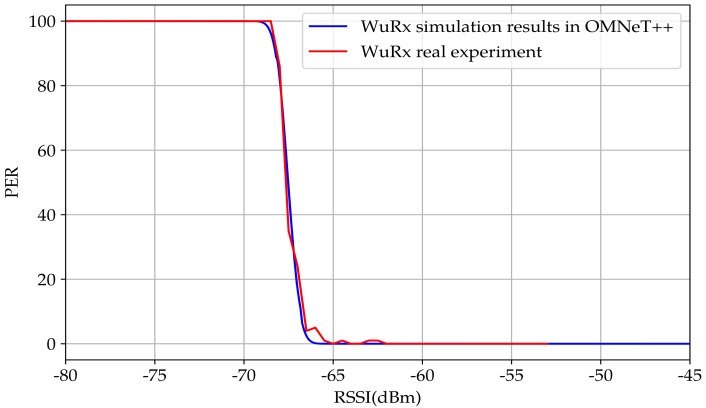
Simulation of WuRx Based on AS3933 LFPM PER model in OMNeT++ discrete event simulator.

**Figure 20 sensors-23-02394-f020:**
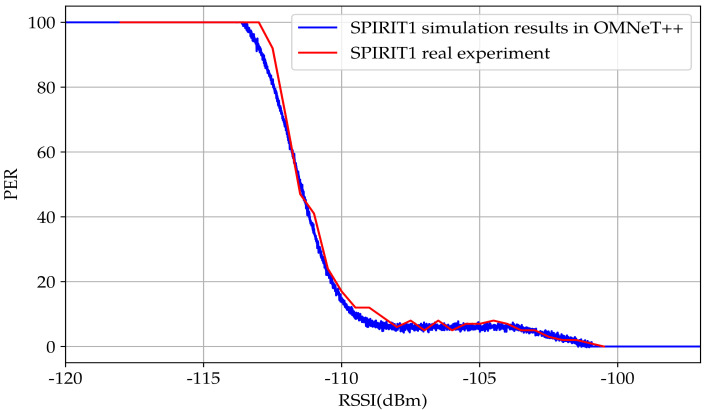
Simulation of Spirit1 transceiver PER model in OMNeT++ discrete event simulator.

**Figure 21 sensors-23-02394-f021:**
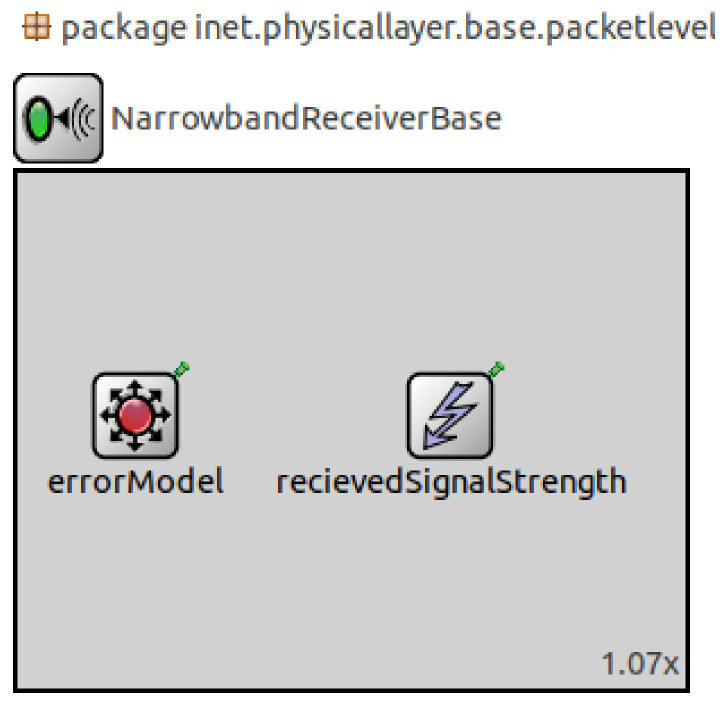
RSSI network description (NED) module.

**Figure 22 sensors-23-02394-f022:**
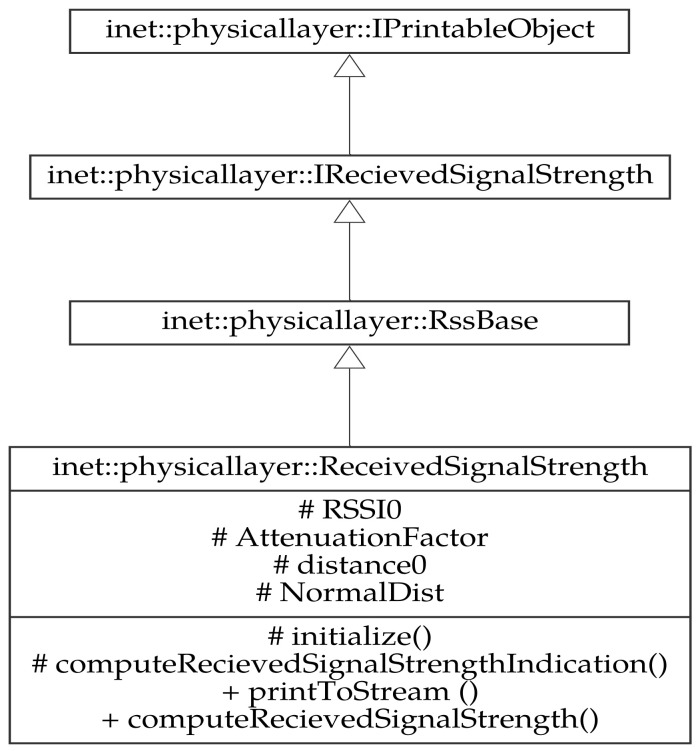
RSSI module in OMNeT++ discrete event simulator.

**Figure 23 sensors-23-02394-f023:**
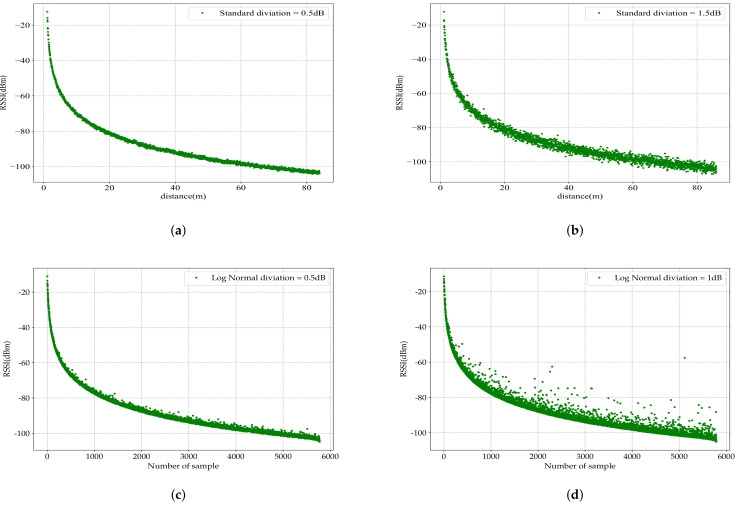
Results of the simulation of the RSSI with different standard deviation values: (**a**) with low noise levell (**b**) with high noise figure level; (**c**) using the log-normal distribution with low deviation; (**d**) using the log-normal distribution with higher deviation.

**Table 1 sensors-23-02394-t001:** Comparison of different simulators of wireless sensor networks.

Criterion	TOSSIM	NS2	OMNeT++/INET
**Nature of simulator**	Emulator	Simulator	Simulator
**Type of the simulator**	Discrete-event	Discrete-event	Discrete-event
**License**	BSD ^1^-license	GNU GPLv2 ^2^ license	Academic Public License. INET models under LGPL ^3^ or GPL.
**User Interface**	GUI ^4^: through TinyViz. Supported languages: Python, C++, and NesC	GUI: through Nam. Supported languages: C++, and OTcl	GUI: a built-in GUI is available Supported languages: C++, and NED ^5^
**Supported platforms**	Linux and Windows	Linux, MacOs, and FreeBSD	Windows, Linux and Mac OSX
**Heterogeneity**	No	No	Yes
**modeling**	Available	Available	Available
**Mobility model**	Yes	Yes	Yes
**Wireless medium model**	Path loss models: lognormal shadowing	Path loss models: shadowing, 2-ray ground, free space	Path loss models: free-space, log-normal shadowing, Rayleigh fading, 2-ray ground, rician fading, Nagasaki fading
**Other models**	Noise modeling	-	Background noise, obstacle loss, and propagation models
**Battery Model**	No	Only for ideal	Yes
**Energy harvester model**	No	No	Yes
**RF ^6^ states**	Yes	Yes	Yes
**Limitation**	Cannot model energy harvester units	Cannot model sensing or processing units	Model expects all parameters and gates to be in place and network fully built when initiating it

^1^ Berkeley Source Distribution. ^2^ General public license version 2. ^3^ Lesser general public license. ^4^ Graphical user interface. ^5^ Network description. ^6^ Radio frequency.

**Table 2 sensors-23-02394-t002:** Configuration parameters of the ADAURA radio frequency digital attenuator.

Specifications	Value
**Attenuation step size (dB)**	0.5
**Operating frequency (MHz)**	868
**WuRx attenuation range (dB)**	from 53 to 80
**SPIRIT1 attenuation range (dB)**	from 62 to 80
**SPIRIT1 Tx power using WuRx (dBm)**	11
**SPIRIT1 Tx power without using WuRx (dBm)**	−34
**Impedance (Ω)**	50

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
