# Peer review of "Modeling of Packet Error Rate Distribution Based on Received Signal Strength Indications in OMNeT++ for Wake-Up Receivers"

_sensors, 2023, doi:10.3390/s23052394_

Round 1
Reviewer 1 Report
1) The manuscript novelty compared to the existing works is Not clear at all. This has to be addressed in the abstract and introduction sections.
2) Fix typos, e.g., see line 1
3) Write main contributions in bullets in the introduction section. Also, list the main advantages and disadvantages
4) Compile summary and conclusion sections into one short and solid section.
5) Update the references, most of them are outdated.
6) Run additional simulations to show how RSSI changes in fading channels.
Author Response
Thank you for your valuable remarks, some highlights and comparison to the related works have been added to the introduction and abstract.
Many English mistakes and grammar issues have been checked.
Regarding the summary, it have been shortened and modified in a way that give a general overview for the whole proposed work, the conclusion is dedicated to a general glossary and opening new horizons for future work and perspectives.
New updated references have been added to the manuscript and modify the old ones.
More simulation has been performed with different noise distributions to show how RSSI changes in fading channels in section 4.2.

Reviewer 2 Report
Dear Authors
Your research is interesting, complete (you have considered real data and simulations), and well-organized. You start with an introduction about the wake-up receiver (WuRx), highlighting the importance of this energy-saving mechanism (power consumption). You also make a technical comparison about the leading WSN simulators, this is a significant contribution when selecting a simulation tool. Hardware and software metrics have been discussed to be integrated into the objective modular network testbed in C++ (OMNeT++) discrete event simulator afterward, as the received signal strength indicator (RSSI) for the hardware metric case and the packet error rate (PER) for the software metric study case using the AS3933 basedWuRx and Spirit1 Low data rate, low power Sub 1GHz transceiver, the different behavior of the two chips is modeled using machine learning (ML) algorithms to define the parameters for both radio modules.
My suggestions:
- In the simulation section, they should indicate more explicitly the execution parameters.
- It would also be good to do experiments (simulations) considering as a variable the number of nodes and their scalability because this also affects the power consumption of the WSN.
- You must explain in more detail the process and results of ML algorithms used, to define the parameters for both radio modules.
Author Response
Thank you for your valuable remarks, the execution parameters are more highlighted and detailed.
Many English mistakes and grammar issues have been checked.
More graphs for the ML analysis are added, such as the QQ-Plot in section 3.2, to clarify more about the data analysis process.
A revision has been made, to include the definition of parameters that uses the non-linear regression used by the mathematical function erf().
More Routing algorithms and scalability for Energy Neutral Operation would be added in future work in the WuRx model in the simulator, since we concentrate on the construction of the WuRx unit for this moment.

Reviewer 3 Report
This paper focuses on the modeling or integration of the link quality estimator of WuRx technology in the simulator, which can effectively reduce the energy consumption of the system. The research direction is of great significance to the development of long-term monitoring and embedded applications. However, the paper needs to make some changes before receiving publication. My detailed comments are as follows :
1. Does the simulation system involve input parameters ? How are they identified or collected ? Please briefly explain.
2. Several algorithms have been mentioned many times, which can be briefly explained in the paper, such as ML algorithm based on supervised learning, RSSI-based ranging algorithm, etc.
3. The OMNeT++discrete event simulator mentioned in this paper has some limitations. Can you specify its limitations and improvement methods ?
4. Please confirm that Section 2.4 contains only one sub-section. Integrating Section 2.4.1 into Section 2.4 or splitting it will make the article look more concise and easy for readers to understand.
5. Two papers about network routing should be cited:
[1] Traffic dynamics on multilayer networks with different speeds. IEEE Transactions on Circuits and Systems II: Express Briefs. 2022, 69(3): 1697 - 1701.
[2] An improved optimal routing strategy on scale-free networks. IEEE Transactions on Circuits and Systems II: Express Briefs. 2022, 69(11): 4578 - 4582.
6. The fifth section briefly summarizes the work of the previous section, which makes the paper seem slightly redundant and repeats with the end of the introduction of the first section. I suggest you rearrange the structure of the paper.
7. What causes the attenuation in the simulation, what factors are simulated in the real environment, and how to control the attenuation? Please answer briefly.
Author Response
I appreciate your time and efforts for the valuable provided remarks
Firstly, The simulation system involves different parameters that can be configured by the user in the INI file of the OMNeT++ simulator.
Such as the transition interval and the sensitivity for the PER module (see Figure 18) and also for the RSSI module that integrate the attenuation factor for example, which could cause the attenuation in the simulation.
Besides other parameters like the distribution, distance reference, and RSSI reference values(see Figure23).
Secondly, more simulations have been performed with different noise distributions to show how RSSI changes in fading channels in section 4.2.
More graphs for the ML analysis are added, such as the QQ-Plot in section 3.2 to clarify more about the data analysis process.
A revision has been made to include the definition of parameters that uses the non-linear regression used by the mathematical function erf() and to briefly explain some mentioned algorithms, such as the RSSI-based ranging method.
Considering the OMNeT++ discrete event simulator limitations, it lacks observing of some metrics, especially for new WSN protocols such as the WuRx.
One of the objective of this work is the integration of those metrics after an exhaustive measurement.
Regarding your remark of the organization of the section 2.4. It has been already corrected and sets it right.
The paper bibliography has been updated, many recent references have been mentioned among them the two mentioned publications, which I found them very interesting and related to the work objectives. The ideas also can be deployed for More Routing algorithms and scalability for Energy Neutral Operation in future work in the WuRx model in the simulator.
Finally, regarding the summary, it has been shortened and modified in a way that gives a general overview of the whole proposed work, the conclusion is dedicated to a general glossary and opens new horizons for future work and perspectives.

Reviewer 4 Report
The author presents in the Introduction section information about how WuRx and MUC work. But, I suggest this information and figures be in another part, out of the introduction.
Provide information about frequency used in experiments, including about, low power Sub 1GHz transceiver and 868 MHz transceiver RF-WuRx. Is it different frequencies?
Such as shown in fig. 9, the PER is affected by dB attenuation where from 55 to 65dB is low and for up to 68dB the PER is almost 100%. I recommend only using information about PER for 60 to 70dB, cause the variation of PER is better shown in this range.
Figure 13, it was considered other distributions for modeling this PERxRSSI graph, such as exponential.
Figure 14 was considered a Weibull distribution. Weibull is a heavy-tailed distribution. How it was defined values of the parameters of that? About heavy-tailed distributions provide information about the use of others such as Pareto, Pareto type II, Log-normal, and log-Cauchy.
To validate the use of Weibull, the author must create a sample of data using parameters carried out, and then compare it with different real data samples. With this, compare data from real samples and simulated samples with a QQ-Plot adherence.
The author suggests an in the deep analysis of discuss the investigation of different link quality metrics, both hardware 11 and software metrics are going to be discussed to be integrated into the objective modular network 12 testbeds in C++. However, Figures 16 and 21 it is showed UML descriptions. UML it not relevant to inform this paper, cause It is not a target for this paper to suggest a specific software to readers.
In general, the paper provides practical tests in real scenarios for WuRx system in WSN. But, does not bring any newest contributions. The main question that authors must answer is "Different other mechanisms and techniques available in the literature, how can be improved the process of WuRx keep a low-cost consumption?" I suggest this paper can be submitted for practice journals.
Author Response
Thank you for your remarks, some parts and clarifications have been added to the introduction in order to highlight better the paper objectives. The detailed WuRx definition with the architecture description are kept in the introduction for the balance of the structure with the other sections.
Regarding the Fig.9 (Fig.10 in the updated manuscript), the PER values in the X axes were eliminated in the plot out of the transition interval region.
Figure 18 and 23 are both updated, describes the hierarchical of the module used in the OMNET++ this explain more the included parameters that searcher could initiate during the simulation for adopting more scenarios in the future.
Both radios (SPIRIT1 transceiver and WuRx) are operating in the 868MHz central frequency, but a specific data rate is considered for matching the AS3933 central operating frequency which is 18.72KHz with a specific address.
The technical ward for the SPIRIT1 transceiver is “Low data rate, low power Sub 1GHz transceiver” we use it sometimes to avoid repetition but it is still confusing, so we update the text to avoid such mistakes.
QQ-Plot is added for the proposed theoretical model of the PER in the SPIRIT1 transceiver and the WuRx(Fig. 15 and 16), and as shown the set of experimental data in the transition interval is not very intensive and we needed such a distribution with multiple parameters to consider the minimum OLS taking in consideration also the OMNeT++ module integration. So I found that the Weibull and the log-Cauchy distribution are very considerable but the log-Cauchy is not very applicable in the constraint of integration.
The modeling of the PERxRSSI modules in OMNeT++ provides the physical sensing element that uses a physical interpretation,
based on mathematical functions. The model prediction could be used in many power management algorithm, since the link quality plays an important role in the reliability and durability of the network. The realization of a realistic emulation could be achieved, with such research, and a lot of studies could be attained in the new green energy technologies, when manipulating the real-world WuRx behavior, many routing protocols could be used and tested according to the scalability of the network.

Round 2
Reviewer 1 Report
thank you
Reviewer 2 Report
Dear Authors
It is much better.
Reviewer 3 Report
I am glad to see the revised version. The paper has been improved and the comments have been addressed properly. I now recommend the acceptance of the paper.
Reviewer 4 Report
Thank you for your comments.